# The Effect of Scaling, Retrieval Augmentation and Form on the Factual Consistency of Language Models

**Lovisa Hagström**[1]  **Denitsa Saynova**[1]  **Tobias Norlund**[1]
**Moa Johansson**[1]  **Richard Johansson**[1,2]

[1] Chalmers University of Technology    [2] University of Gothenburg
{lovhag, saynova}@chalmers.se

## Abstract

Large Language Models (LLMs) make natural interfaces to factual knowledge, but their usefulness is limited by their tendency to deliver inconsistent answers to semantically equivalent questions. For example, a model might predict both "Anne Redpath passed away in *Edinburgh*." and "Anne Redpath's life ended in *London*." In this work, we identify potential causes of inconsistency and evaluate the effectiveness of two mitigation strategies: up-scaling and augmenting the LM with a retrieval corpus. Our results on the LLaMA and Atlas models show that both strategies reduce inconsistency while retrieval augmentation is considerably more efficient. We further consider and disentangle the consistency contributions of different components of Atlas. For all LMs evaluated we find that syntactical form and other evaluation task artifacts impact consistency. Taken together, our results provide a better understanding of the factors affecting the factual consistency of language models.

## 1 Introduction

We have recently observed the development of several highly performant pretrained large language models (LLMs) that have expanded the boundaries of what we can expect from foundational language models. These recent successes have highlighted the potential of using language models as a simpler interface to factual knowledge (Petroni et al., 2019; Dinan et al., 2019).

However, in fact critical settings we require not only high accuracy but also consistency, i.e. robustness to lexical variations in semantically equivalent queries. Recent LM developments have mainly improved on accuracy, while the question of consistency has seen less attention. As exemplified in Figure 1, even SoTA LMs may produce different outputs depending on lexical variations in semantically equivalent queries (Elazar et al., 2021; Ribeiro et al., 2019; Cao et al., 2021). This sug-

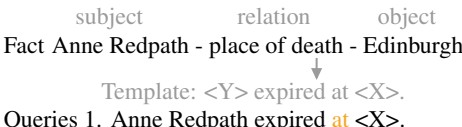

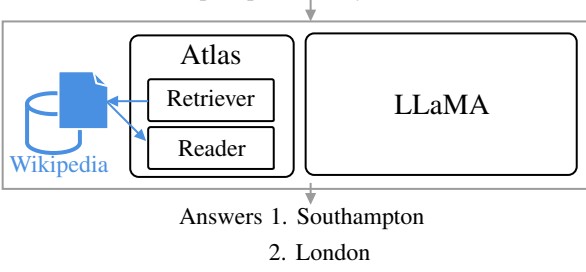

Figure 1: Overview of how consistency is computed in ParaRel for Atlas and LLaMA.

gests that performance on benchmarks for factual information does not measure true factual knowledge, since models do not generalize well across rephrased prompts.

Problems with inconsistency extend beyond fact-related accuracy, as they also indicate a tendency to hallucinate and a lack of factuality in foundational LMs (Ji et al., 2023). As inconsistent models may be fragile in unexpected manners, they are difficult to interpret and we are in a worse position for providing limitations on what they may generate (Wang et al., 2022).

To promote desirable properties such as robustness and interpretability we may need to reconsider our model designs. As an alternative to up-scaling, we have seen the rise of model designs guided by inductive biases to promote various properties (Feder et al., 2022). Examples of such models are text retrieval-augmented models that condition predictions on retrieved text passages for improved adaptability, interpretability and efficiency (Izacard and Grave, 2021; Izacard et al., 2023; Wu et al., 2022), neurosymbolic models that condition predic-

tions on logic for reasoning capabilities (Pacheco and Goldwasser, 2021) and fact-injected LMs for interpretability (Verga et al., 2021).

Our goal in this paper is to improve our understanding of consistency in foundational LMs. We start by investigating methods to improve model consistency. We consider two potential solutions, 1) model up-scaling and 2) Wikipedia text retrieval augmentation, building on suitable inductive biases related to factual consistency. Furthermore, we identify and investigate potential sources of inconsistency of LMs. Our contributions can be summarized as follows:

- We investigate ParaRel, a benchmark for evaluating consistency (Elazar et al., 2021) and develop an improved version of it[1] (Section 2). We refer to this version as ParaRel* throughout this paper. We have removed ambiguous fact duplicates and added four query-level metrics to estimate inconsistency sources related to evaluation task format, such as linguistic form effects (Section 4).

- We investigate the hypothesis that model scaling mitigates inconsistency (Section 3.1). Our evaluation of varying sizes of the LLaMA and Atlas models (Touvron et al., 2023; Izacard et al., 2023) show that scale improves consistency, but with diminishing returns.

- We hypothesize that retrieval augmentation mitigates inconsistency and investigate this using Atlas (Izacard et al., 2023) – a retrieval-augmented model – along with non-retrieval models of similar scale (Section 3.2). We show that Atlas outperforms all baselines on the ParaRel* task.

- We also investigate causes and correlations related to consistency of Atlas (Section 5).

  - We define metrics for measuring retrieval consistency and find that the Atlas retriever components generally is consistent and that retrieval consistency correlates with prediction consistency.

  - Through interventions on the retrieval component we find that the consistency and relevance of the retrieved result both affect the consistency and accuracy of the predictions.

  - We investigate to what extent the reader component depends on term frequencies in the retrieved result and find it to be more inconsistent when this dependence is weaker.

## 2 Data

We describe the original ParaRel benchmark in Section 2.1 and our improvements in Section 2.2. Section 2.3 explains the metrics studied for ParaRel*.

### 2.1 ParaRel

ParaRel (Elazar et al., 2021) is based on LAMA (Petroni et al., 2019), an evaluation task based on Wikidata that measures factual knowledge stored in LMs through prompting for subject-relation-object tuples. ParaRel adds a layer of semantically equivalent cloze-style prompts to LAMA, which in turn allows us to measure the consistency of LMs with respect to the knowledge tuples represented by LAMA (see Figure 1). The idea is that a model is consistent if it is invariant to query paraphrases.

ParaRel measures *consistency* for N-1 relations, for which there is only one correct object alternative for a subject-relation pair, and *plausibility* for N-M relations, where there are several correct objects for a subject-relation pair. We only consider the N-1 relations that measure consistency. Additionally, following Elazar et al. (2021), we simplify the task by restricting the candidate sets. The models are only allowed to generate an answer from the alternatives in the ParaRel* data for each relation.

### 2.2 Analysis and Improvements

Analysis of the ParaRel data showed that it contains exact duplicate tuples – e.g. *Audi R8 produced-by Audi*. This refers once to the Wikidata entity *Q758775* (prototype race car) and once to *Q758778* (sports car model), which are not distinguishable by name only. Additionally, duplicated subject-relation pairs (with a different object) also exist in the data. For example, *SNES-CD produced-by Sony* and *SNES-CD produced-by Nintendo* are both present in the data. Appendix F presents statistics for the number of such duplicates.

Since we aim to use ParaRel to measure consistency for N-1 relations, we remove all subject-relation instances that occur more than once in the dataset. Relation P37 *official-language* had 280 duplicates out of 900 data entries and cannot be considered to be a N-1 relation, why we completely remove it from our updated ParaRel* version.

---

[1]Code and datasets available at https://github.com/dsaynova/pararel.

This removal of duplicates resulted in an updated ParaRel* of 30 relations with 21,830 data tuples in total, instead of the original 23,097. Some of the retained relations contained more duplicates than others, but never more than 10% of the data tuples.

## 2.3 Evaluation Metrics

ParaRel provides several statistics that can be used to evaluate model performance. We mainly focus on three: *Consistency* – pairwise agreement between prompts for each tuple; *Accuracy* – accuracy when using the original LAMA prompt; *Consistent & Accurate* – what percentage of subjects get assigned the correct object for all prompts.

The consistency metric results in measurements for all possible query pairs per tuple and relation. To get one consistency value per model evaluated, we calculate the micro-average of the consistency values across tuples per relation and then take the macro-average across relations, following Elazar et al. (2021). The pairwise comparisons used to estimate consistency imply that other metrics relating to consistency, by correlation or stratification, also need to be on a pairwise query-level.

## 3 Effect of Scaling and Retrieval Augmentation

To investigate the effect of scaling and retrieval augmentation on the consistency of LMs we evaluate LLaMA (Section 3.1) and Atlas (Section 3.2) on ParaRel* and report the results in Section 3.3.

## 3.1 Effect of Scaling

Model scaling has been a successful approach to improving performance on many NLP tasks. We evaluate LLaMA (7B, 13B, 33B and 65B parameters) (Touvron et al., 2023) on ParaRel* (see Appendix A for details on how we do this) and measure whether consistency improves with model size. LLaMA represents traditional state-of-the-art LLMs, being decoder-only auto-regressive LMs trained on over a trillion tokens. We also investigate this for retrieval-augmented models by comparing two sizes of the Atlas model.

## 3.2 Effect of Retrieval Augmentation

We expect retrieval-augmented LMs to generally be more consistent than standard LMs. Given a consistent retrieval component, the prediction of a retrieval-augmented model is conditioned on something bound to be more consistent than the query

alone. Retrieval augmentation from e.g. Wikipedia has been successful for several fact-intensive question answering tasks (Izacard et al., 2023; Lewis et al., 2020) and reduced hallucination (Shuster et al., 2021; Thoppilan et al., 2022).

Atlas is a retrieval-augmented LM optimized for few-shot knowledge-intensive tasks. It was developed by Izacard et al. (2023) and retrieves passages from Wikipedia and a common crawl dump. The model consists of a dense *retriever* based on the Contriever architecture (Izacard et al., 2022) and a *reader* based on the T5 seq2seq architecture (Raffel et al., 2020). The model performs on par with, or even better than, many larger fully-parametric language models on e.g. Natural Questions (Kwiatkowski et al., 2019).

We evaluate Atlas on ParaRel* and compare its performance to relevant baselines, i.e. non-retrieval-augmented model counterparts. We mainly use the base version of Atlas with 330M parameters in our analysis and to some extent also the Atlas-large with 880M parameters, using the model weights released by the authors.[2] We use Wikipedia 2017 passages as the retrieval corpus since this should match the temporal origin of the ParaRel* data and keep the retrieval indices fixed. We retrieve 20 passages per query.

Since Atlas has been pre-trained with MLM as the pretext task, we can evaluate it zero-shot with some adaptations (see Appendix A).

**Baselines** To properly investigate the effects of the different model development choices for Atlas on consistency, we need to compare Atlas to reasonable baselines. To reason about the benefits of the retrieval augmentation and additional training provided for Atlas, we compare against the fully-parametric T5-base model Atlas was initialized from.[3] To assess the benefits of retrieval augmentation specifically, we evaluate Atlas in a *closed-book* setting without augmentation, while we acknowledge that the model has not been adapted to this setting.

We also compare Atlas-base and BERT-large to estimate the benefits of retrieval augmentation. The latter model should be a good representative for the retrieval-free setup as it has an advantage in all aspects except for retrieval augmentation. BERT-large has slightly more parameters than Atlas-base, is better adapted to the encoder-based ParaRel* task

---

[2] https://github.com/facebookresearch/atlas#models
[3] google/t5-base-lm-adapt

| Model | Cons | Acc | C & A |
|-------|------|-----|-------|
| atlas-base | 0.74±0.15 | 0.80±0.16 | 0.42±0.27 |
| atlas-large | 0.77±0.14 | 0.81±0.14 | 0.47±0.29 |
| atlas-base* | 0.60±0.23 | 0.41±0.24 | 0.24±0.22 |
| t5-base | 0.59±0.23 | 0.39±0.23 | 0.22±0.19 |
| bert-base | 0.58±0.24 | 0.46±0.26 | 0.27±0.24 |
| bert-large | 0.60±0.23 | 0.48±0.26 | 0.29±0.27 |
| llama-7b | 0.67±0.18 | 0.67±0.21 | 0.36±0.28 |
| llama-13b | 0.68±0.17 | 0.70±0.21 | 0.39±0.28 |
| llama-30b | 0.70±0.17 | 0.75±0.18 | 0.42±0.28 |
| llama-65b | 0.71±0.16 | 0.75±0.19 | 0.43±0.28 |

Table 1: ParaRel* results of Atlas, LLaMA and baselines averaged over all 30 N-1 relations. *=closed-book.

and has mainly been trained on Wikipedia, which could give it an advantage (Elazar et al., 2021).

### 3.3 Consistency Performance

Table 1 shows the ParaRel* zero-shot results for all evaluated models for all 30 relations. Following the work by Elazar et al. (2021), all tables report the macro-average and standard deviation over the different relations. Our findings based on these results are as follows:

**Retrieval augmentation has a sizeable effect on consistency and accuracy**   We find that the Atlas models are more consistent compared to all other evaluated models. Since Atlas-base outperforms BERT-large in spite of the disadvantages outlined in Section 3.2, we have good reason to believe that retrieval augmentation leads to improved consistency and is the main contributor to the superior performance of Atlas. Atlas-base and large are also more accurate on ParaRel*. Accuracy is not the main interest of this work, however we expect these effects to be interconnected.

While Atlas has superior consistency on ParaRel* compared to the other models investigated, it does not achieve *perfect* consistency. Despite being developed for fact-critical tasks and allowed to retrieve information from Wikipedia, the model can generate different answers to semantically equivalent factual questions. We investigate potential reasons related to the evaluation task format and reader-retriever interaction in Section 4 and Section 5.

**Model scaling has a sizeable effect on consistency and accuracy**   We observe improved consistency and accuracy performance with size for both LLaMA and Atlas. The consistency increases with approximately 1% in LLaMA and 3% in Atlas per doubling in model size. Similar effects of up-scaling were also observed by Elazar et al. (2021). The larger LLaMA models also outperform BERT-large, while they do not outperform Atlas.

**Retrieval augmentation is more efficient than up-scaling in increasing consistency**   Atlas-base (330M parameters) performs on par with LLaMA-65B despite being 90 times smaller. Evidently, retrieval augmentation with Atlas is more efficient than up-scaling for increased model consistency. We also note that up-scaling LLaMA yields diminishing returns with respect to consistency. Consequently, we have found a foundational NLP issue – consistency – for which model up-scaling is not the complete solution. Using model designs guided by inductive biases is a promising approach for robust and interpretable NLP systems.

## 4 Effect of Evaluation Task Format

Apart from retrieval-augmentation and scaling, we also expect form and evaluation task format to have had an impact on the consistency results in Table 1. ParaRel*, like many other fact-critical evaluation sets (Petroni et al., 2019; Norlund et al., 2021; Sciavolino et al., 2021), relies on automatic prompt generation by substituting different subjects into pre-defined templates. Additionally, both templates and testing data are extracted and constructed semi-automatically. Using synthetic data and automated steps in benchmarks generation is common in NLP pipelines (He et al., 2022; Meng et al., 2022b). However, the convenience of this approach comes with the risk of producing disfluent queries. Previous work has found that language models learn form and syntax faster than semantics and general natural language understanding (NLU) (Zhang et al., 2021). We hypothesize that this can lead to a prioritization of text fluency and syntax over facts, influencing model consistency. Therefore, we also evaluate the sensitivity of our models to issues that may arise from the evaluation task format.

### 4.1 Metrics to Estimate Inconsistency Sources from the Evaluation Task Format

Issues from the evaluation task format manifest as four query-related effects, falling into three groups: semantic overlap in answers, unidiomatic language (on both template and object level), and subject-object similarity. While semantic overlap is caused

by the evaluation setup, the remaining issues relate to a larger issue of the effect of language form on model behaviour.

**Semantic overlap**  The exact string matching used in ParaRel* to determine model success, taken together with the constrained decoding may lead to biased estimations of model performance. Some relations include more ambiguity than others, where the model is allowed to choose between semantically close answer alternatives, but only one of these is accepted as a correct answer. Relation P30 *located-in-continent*, for example, only includes *Africa*, *Americas*, *Antarctica*, *Asia*, and *Europe*, while relation P101 *field-of-work* contains both *biology* and *science*, and relation P19 *born-in* contains both *Glasgow* and *Scotland*. We hypothesise that this added ambiguity may affect consistency as measured by ParaRel*, and therefore one of the comparisons we perform is on the consistency of the 12 relations that have semantic overlap in the answer options (for a full list see Appendix H) and the remaining 18 relations that do not have an overlap.

**Unidiomatic language**  Certain template-answer combinations in ParaRel* result in disfluencies. This is caused by either an ill-fitting template or an object, that, when combined with a template, results in unidiomatic language.

For example "Anne Redpath died in *Edinburgh*" is much more natural than "Anne Redpath died at *Edinburgh*", even though to a human the meaning is clear in both. This affects templates in 6 relations (for a full list of unidiomatic templates see Appendix H).

Furthermore, "Solar Mass is named after *the Sun*" sounds more natural than the ParaRel* entry "Solar Mass is named after *Sun*". This mainly affects objects in three relations: P101 *specializes-in*, P138 *named-after*, and P361 *part-of*. A manual annotation of the answer alternatives can be found in Appendix H, where we identified objects that were expressed unnaturally within the template, typically because they lacked a determiner or plural form. A more systematic classification to refine these labels through larger-scale human annotation or processing with a grammar checker is left for future work.

To evaluate the effect of these issues, we compare the pairwise consistency of affected relations between the cases with and without unidiomatic

objects and templates respectively.

**Subject and object similarity**  Several relations contain an overlap between the subject and object – for example – *Nokia N9 produced-by Nokia*. We hypothesise that in these cases the model can be guided by a simple heuristic rather than accessing factual information, which may lead to an increased performance on ParaRel*. We identify 9 relations that contain more than 20% of tuples with an overlap between subject and object stems (see Appendix H) and report pairwise consistency for queries with and without subject-object similarity.

## 4.2 Evaluation task format has an effect on consistency

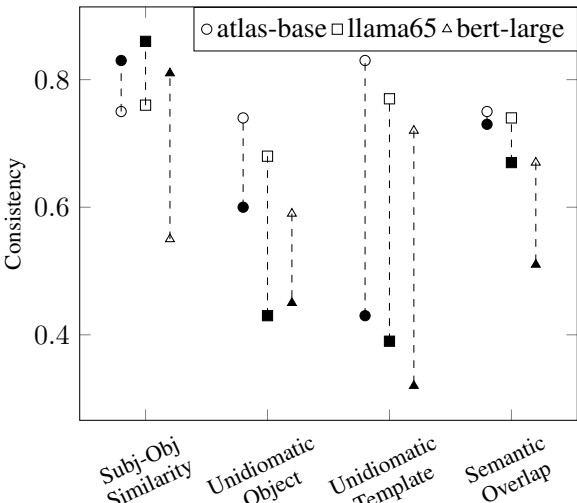

Figure 2: Performance for query-related sources of inconsistency. Filled-in marks indicate performance on affected data, non-filled-in – on data not affected.

Figure 2 indicates the consistency performance and analysis of model sensitivity to evaluation task format issues, including form. Detailed results are available in Appendix G. All evaluation task format related phenomena discussed have an effect on consistency across all model settings. We see that unidiomatic objects and templates affect all models on a similar scale, whereas some models are less susceptible than others to subject-object similarity and semantic overlap in the answer alternatives.

Our results show that all evaluated LMs are fragile to language usage, such that we cannot expect them to be robust in the face of less fluent queries. Judging whether this is acceptable LM behaviour could be based on the severity of the disfluency. We could leverage our results to create *better* evaluation tasks by filtering out samples that prevent

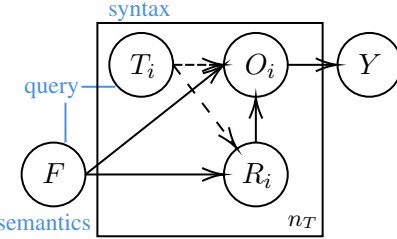

Figure 3: Causal effects on model consistency for a retrieval-augmented model. $O_i$ denotes the prediction made by the model given retrieved information $R_i$ and query, based on a template $T_i$ and invariant fact $F$. For each fact we have $n_T$ templates. $Y$ is the consistency, i.e. fraction of equal $O_i, O_j$ pairs. Dashed lines indicate weak effects.

us from measuring our quantity of interest, such as factual knowledge under correct language usage. However, we argue that in the face of the less severe linguistic variations measured in this work, it is desirable to have a LM that is robust and stays faithful to its factual predictions.

## 5 Effect of Retrieval Components

We found Atlas to be more consistent compared to LMs without retrieval augmentation. In this section we perform a deeper analysis on the effects of letting a model retrieve over some corpus with focus on the retriever and reader components.

We use the DAG depicted in Figure 3 as a starting point for our reasoning. We define a query to be the combination of a fact tuple, $F$ and template $T$, as depicted in Figure 1. We can see $F$ and $T$ as the semantics and syntax of a query respectively. A model prediction for a certain query and retrieved information can thus be described as $O(T, F, R(F, T|\beta)|\theta)$, where $\theta$ and $\beta$ are model parameters of the reader and retriever respectively.

The motivation for why retrieval-augmented LMs should be more consistent than standard LMs builds on two parts related to the retrieval and inference steps of the retrieval-augmented model. First, a perfect retrieval result should ideally be independent of the syntax (template) of a query conditioned on the semantics (facts) of the query. The retriever is trained on the more narrow task of identifying relevant factual aspects of a query and matching those to the information in the retrieval corpus, thereby becoming less dependent on syntactic variations. Furthermore, the retriever has a more restricted output space and can only choose between already existing facts rather than create new ones. Expres-

sion (1) summarizes the expected behavior of an ideal retriever.

$$R(F, T|\beta) \perp T|F \qquad (1)$$

Second, the generated output of an ideal retrieval-augmented model for a fact-related query should be independent of the syntax, given the semantics of the query and corresponding perfect retrieved information. Expression (2) formalizes this idea. In the ideal case, the retrieved information adds an inductive bias to the generation process that conditions the prediction to be independent of the syntax of the query.

$$O(T, F, R|\theta) \perp T|R, F \qquad (2)$$

In practice, we do not expect Atlas to correspond perfectly to the ideal case. Our experiments investigate these dependencies and how they interact with the consistency. We measure retriever consistency (Section 5.1), the effect of the retrieved information on consistency (Section 5.2) and the dependence of the reader on the retrieved information (Section 5.3).

### 5.1 Retriever consistency

We hypothesize that the retriever is less dependent on syntax compared to a standard LM and thereby more consistent. To estimate the dependence on syntax of the Atlas retriever, we measure retriever consistency by estimating the pairwise retriever agreement for the prompt pairs for each tuple in ParaRel*. As no automatic method exists for estimating retriever consistency, we propose and evaluate three novel metrics. For Atlas, the retrieved information is in the format of 20 Wikipedia passages, so our metrics for pairwise retriever agreement rely on 1) id overlap between retrieved passage sets, 2) title overlap between retrieved passage sets and 3) retriever query embedding similarity.

For the two first metrics, we calculate the id and title overlap of the 20 retrieved documents (normalized to 1), estimating exact passage match and whether passages were retrieved from the same Wikipedia page respectively. These metrics do not account for the fact that different passages, coming from different Wikipedia pages, could contain the same information, especially in the current setting that queries for simple tuple-based facts. Thus, the metrics are strict and should give a lower bound on retriever consistency.

The embedding based retriever consistency metric is more soft. For this metric, we consider the embeddings produced by the retriever for each query paraphrase and estimate their similarity. Since the Atlas retrieval is based on a search with respect to the similarity between embeddings of queries and passages, the query embedding similarities for paraphrases can be expected to reflect retriever consistency. We use the cosine similarity between retriever embeddings for paraphrase pairs as an estimate for retriever consistency.

To get a baseline performance of the retriever consistency metrics we also evaluate them on pairs of randomly sampled passages previously retrieved by Atlas-base. We use two methods for random sampling, 1) sample passage/query pairs completely at random across relations and subjects, denoted as *r-all*, and 2) sample passage/query pairs from retrieval results for the same relations but different subjects, denoted as *r-subject*.

| Model | Metric | Similarity $\mu$ | Similarity $\sigma$ |
|---|---|---|---|
| r-all | *id* | 0.00±0.00 | 0.01±0.01 |
| | *title* | 0.00±0.00 | 0.02±0.01 |
| | *emb* | 0.54±0.04 | 0.06±0.01 |
| r-subject | *id* | 0.01±0.01 | 0.02±0.02 |
| | *title* | 0.01±0.01 | 0.02±0.02 |
| | *emb* | 0.64±0.04 | 0.06±0.01 |
| atlas-base | *id* | 0.51±0.10 | 0.19±0.03 |
| | *title* | 0.58±0.10 | 0.20±0.04 |
| | *emb* | 0.88±0.04 | 0.05±0.02 |
| atlas-large | *id* | 0.52±0.10 | 0.19±0.03 |
| | *title* | 0.59±0.09 | 0.20±0.04 |
| | *emb* | 0.88±0.04 | 0.05±0.02 |

Table 2: Similarity metrics used to estimate retriever consistency. *r-all* denotes random sampling over subjects and relations, and *r-subject* sampling over subjects with the relation fixed. $\mu$ refers to the distribution of the mean per relation and $\sigma$ refers to the distribution of the standard deviation of the metrics per relation.

We apply our retriever consistency metrics to the Atlas retriever and report the results in Table 2. Our findings are as follows:

**The Atlas retriever exhibits a weak dependency on syntax** For the id and title overlap metrics, we first set a threshold in order to label a retrieval result as consistent. From inspection by example we found the retriever able to find many passages on the same topic, some with semantic overlap despite not sharing the same id or title. An average overlap of 0.5 could thus be seen as a quite consistent performance, especially if we compare them to the low values for the baselines. There are multiple factors that can account for the incomplete overlap – e.g. the number of paragraphs that are related to a particular fact can vary.

For the metric based on embedding similarities, we observe a significant increase in embedding similarity for queries with the same semantic meaning compared to for random pairs of queries. The retriever embedding similarities also capture some relation information, as the embeddings for random queries sampled over subjects have an increased similarity compared to the random baseline.

We conclude that the Atlas retriever is quite consistent according to our metrics, especially if we compare against the randomized baselines. However, its consistency is not perfect across paraphrases, indicating a weak dependence on syntax. We note that there are many nuances to automatic metrics for retriever consistency that remain to be explored. Future work could for example include measuring the consistency among passages retrieved for the *same* query.

**Reader and retriever consistencies are correlated** To estimate the effect of the retriever consistency on the overall model consistency, we measure the Pearson correlations between reader consistency and our retriever consistency metrics. We expect consistent predictions to correlate with consistent retrieval results and vice versa, building on arguments related to Expressions (1)-(2).

We observe a weak correlation between all retriever consistency metrics and the Atlas consistency. For both id and title the Pearson correlation is $0.16 \pm 0.08$, while for query embedding similarity it's $0.14 \pm 0.13$. As previously discussed, we expect retrieval augmentation to condition the prediction to be less syntax dependent, why we expected some form of correlation to exist, but we cannot fully explain why it is not higher. One explanation is imperfect metrics for retriever consistency. Most likely, there are additional factors at play that determine whether Atlas is consistent. After all, the Atlas reader is only conditioned on and not controlled by the retrieved information.

| Intervention | Cons | Acc | C & A |
|---|---|---|---|
| none | 0.74±0.15 | 0.80±0.16 | 0.42±0.27 |
| relevant | 0.79±0.15 | 0.80±0.16 | 0.48±0.28 |
| irr cohesive | 0.61±0.22 | 0.42±0.24 | 0.23±0.23 |
| irr incohesive | 0.58±0.24 | 0.42±0.24 | 0.22±0.22 |

Table 3: Atlas-base results for different interventions on the retrieval augmentation. *relevant* denotes a consistent and relevant retrieved result. *irr cohesive* a consistent and unified result, but irrelevant. *irr incohesive* a consistent but completely random result for each of the 20 passages.

## 5.2 Interventions to further investigate the effect of retrieval augmentation

We carry out intervention experiments on the retrieved information to investigate its effect on Atlas. To test whether consistent retrieved information across paraphrases contributes to more consistent predictions, we set the retrieved information $R_i$ to a fixed value across paraphrases indexed by $i$ in ParaRel*, meaning that $\forall i \, R_i = R$. We then measure the consistency of the reader conditioned on this consistent retrieved information.

Taking the effect of retrieved information quality into account, we test three different approaches for setting the fixed retrieved information for a given fact $F = (s, r, o)$. First, given the corresponding query paraphrases for $F$ indexed by $i$ we take the retrieved passages for one of the paraphrases $i'$ and use that as the retrieved information for the remaining paraphrases $R_{(s,r),i} = R_{(s,r),i'}$, making each $R_{(s,r),i}$ both relevant and consistent. Second, we take the retrieved passages for a query paraphrase $i'$ for another subject $s'$ from the same relation $r$ and set this as the fixed retrieved information $R_{(s,r),i} = R_{(s',r),i'}$, making the retrieved information consistent and cohesive across the retrieved passages but irrelevant. Third, we use a set of completely random passages $R_{\text{rand}}$ as the retrieved information $R_{(s,r),i} = R_{\text{rand}}$, making it consistent but irrelevant and incohesive. See Appendix B for a sketched example. The two latter approaches let us disentangle the effect of relevance from the effect of consistency for the retrieved information.

We report the results from these interventions in Table 3. Our findings based on these results are as follows:

**Consistent and relevant retrieval augmentation causes more consistent predictions** We con-

clude that relevant and consistent retrieved information leads to a more consistent Atlas model, while not perfectly consistent. To some extent, this confirms our hypothesis related to Expression (2), while further investigations are needed to explain the absence of *perfect* consistency. As we saw in our previous results, the effect of form could play a role.

Another potential explanation for the lack of perfect consistency is given by Longpre et al. (2021). They found that a context-augmented LM may ignore the provided context if it contradicts the knowledge stored in the model parameters, hypothesizing that fact *popularity* plays an important role. Future work could examine whether this phenomena also extends to retrieval-augmented models.

**Consistent while irrelevant retrieval augmentation does not result in more consistent predictions** We can also observe from Table 3 how consistent but irrelevant retrieved information not related to the query at hand leads to significant decrease in both consistency and accuracy. We observe similar behaviour regardless of whether the irrelevant retrieved information was cohesive or not (i.e. related to the same fact). Seemingly, the reader prediction is not only dependent on consistent retrieved information, and the reader may be able to discriminate what retrieved information is relevant to the query. This corroborates our theory that additional inconsistency sources for Atlas exist apart from retriever consistency. More assumptions are needed for the variables in Expression (2) before we can accurately describe the workings of Atlas.

## 5.3 Reader dependence on retrieval result

To further reason about the effect of retrieval augmentation on consistency, we would like to know to what extent the Atlas reader is dependent on the retrieval result. We start by investigating one possible heuristic – namely, that the reader relies on frequencies in the retrieved passages.

We estimate the dependence of the Atlas reader on the retrieval result using the rank of the reader prediction according to term frequencies in the retrieved passages. This is further described in Appendix C. Since the Atlas reader may be more or less dependent on the retrieved passages depending on how well it reflects a correct answer to the query we also measure the rank of the gold label following the same approach. We expect a useful result to rank the correct answer higher compared

| Model | Type | Rank | Match | No match |
|-------|------|------|-------|----------|
| base | *pred* | 0.09±0.06 | 0.06±0.04 | 0.19±0.09 |
| | *gold* | 0.07±0.05 | 0.05±0.04 | 0.13±0.07 |
| large | *pred* | 0.08±0.06 | 0.05±0.04 | 0.19±0.08 |
| | *gold* | 0.07±0.05 | 0.05±0.04 | 0.14±0.08 |

Table 4: Frequency-based averaged rankings in the retrieved result of the reader's predictions and the gold labels. The Pearson $\rho$ between consistency, prediction and respectively gold rank were $-0.34 \pm 0.12$ and $-0.18 \pm 0.10$.

to a less useful result.

We expect consistent predictions to be more reliant on the retrieval result compared to inconsistent predictions and can measure this using our defined metrics. Moreover, we expect the reader to be more reliant on useful retrieval results, and thus more consistent for these, compared to for less useful results that are less successful in surfacing the correct answer.

We estimate the dependence of the Atlas reader on the retrieval result using our ranking and report the results in Table 4. An average rank of 0 corresponds to the reader always predicting the top answer candidate according to the retriever. Our findings based on these results are as follows:

**Reader dependence on retrieval result correlates with consistency** The average rank of a prediction is higher for consistent predictions, compared to inconsistent. It could be that the reader for some instances is less faithful to the frequency based signal of the retriever and simply picks an answer less supported by the retrieval result. Drawing from our results in Figure 2, we hypothesize that this could happen if the answer supported by the retrieved information would produce an unidiomatic sentence. Alternatively, some samples may be more difficult for both reader and retriever to process, resulting in lower rankings of the predictions made.

**Retriever frequency of the correct answer correlates with consistency** The retriever generally promotes the correct answer frequency-wise. Additionally, similarly to our previous observation, the retriever is largely better at promoting the correct answer for samples for which the reader is consistent. This indicates that there are samples for which the retriever struggles to find suitable passages and that the reader is less consistent for these.

# 6 Related Work

While there is a large body of work on the factual *accuracy* of LMs, the question of *consistency* has received less attention. This issue was investigated by Elazar et al. (2021), which introduced the ParaRel benchmark which forms the basis of our work here. On the accuracy side, there have been investigations that relate effects of the training data (primarily how frequently facts occur) to the correctness of factual predictions (Mallen et al., 2023; Kandpal et al., 2023; Elazar et al., 2022).

The approach of viewing a complex LM as a causal system and carrying out interventions to investigate the effect of components on the model's behavior was pioneered by Vig et al. (2020). Further work by Meng et al. (2022a) applies this methodology to examining fact associations and proposes a hypothesis of how models recall factual information stored in their parameters.

# 7 Conclusions and Future Work

In this work, we investigate methods to improve factual consistency of LMs and further reason about potential sources of inconsistency. We find that up-scaling, represented by LLaMA, generally improves on consistency, while it is inefficient and surpassed by retrieval augmentation, represented by Atlas. Retrieval augmentation produce the largest consistency improvement. However, Atlas does not achieve perfect consistency even for perfectly consistent retrieved passages, this indicates that additional inconsistency sources are involved and persistent in spite of consistent conditioning. These are potentially related to form anomalies and changeable reader dependency on the retrieved information. In conclusion, we have identified and measured several potential sources of (in)consistency that should aid future work on improving the robustness of LMs, while much remains to be investigated.

Future work includes evaluating consistency of additional model designs, such as context-augmented LLMs and models that build on knowledge graphs (Shu et al., 2022). We could also make use of methods similar to the ones proposed by Meng et al. (2022a) to further investigate parametric versus non-parametric memorization in the context of retrieval augmentation and its effects on consistency.

## Limitations

We reason about benefits related to up-scaling and retrieval augmentation based on our results from Atlas and LLaMA, but this may not give the complete picture of the performance of these model types in general.

Based on the performance of Atlas, we reason about the effect of retrieval augmentation on inconsistency, while we cannot perfectly control for the confounding effect of different training data. Generally, training data has been found to have a big impact on model performance and especially when relating to accessing factual information. The best way to estimate the exact effect of retrieval augmentation would be to use a proper closed-book version of Atlas, trained without augmentation on the same data. However, since the weights for this baseline have not been released we resort to addressing this by marginalization over several other relevant baselines.

Similarly, we compare LLaMA to Atlas to reason about the benefits of LLMs versus retrieval augmentation, while these two models have been trained on different datasets and we cannot marginalize over training data effects. Elazar et al. (2022) found that models trained to a larger extent on Wikipedia performed better on ParaRel, potentially because Wikipedia is a more unified source of factual knowledge. In this sense, LLaMA might be at a disadvantage, due to requiring large amounts of training data that are not guaranteed to be unified or factual.

Furthermore, we base our analysis of consistency on a single benchmark dataset, which to the best of out knowledge is the largest and most relevant dataset for these types of evaluations. However, further investigations into how these results generalize beyond ParaRel* and its specific design choices remains to be investigated.

We focus our study on cloze-style queries, which limits how well the test set can be handled by different models. These prompts are more suited for masked language modelling than for auto-regressive models, so further investigations of the effects of the chosen task formulation is needed.

Furthermore, models cannot express level of certainty, so it is unclear to what extent the performance is tied to the (un)availability of facts in the model. Together with the constrained decoding method applied in this work, this may lead to models being able to surface a fact in certain settings, but that may not serve as a binary distinction between a fact being represented or not in the model. The different level of difficulty of recalling a fact is a further possible confounder left for future work.

## Ethics Statement

Our work has the overall goal of improving consistency of LMs – a goal that, if accomplished, could make AI generated text harder to identify with potential negative consequences. On the positive side, improved consistency of LMs should make them easier to explain and safer to use in fact critical situations. The ParaRel* data used in this work is not associated with any ethical complications.

## Acknowledgements

We are grateful to Marco Kuhlmann, whose support and insightful feedback was essential for the completion of this work.

This work was partially supported by the Wallenberg AI, Autonomous Systems and Software Program (WASP) funded by the Knut and Alice Wallenberg Foundation and by the Wallenberg AI, Autonomous Systems and Software Program – Humanities and Society (WASP-HS) funded by the Marianne and Marcus Wallenberg Foundation and the Marcus and Amalia Wallenberg Foundation. The computations were enabled by resources provided by the National Academic Infrastructure for Supercomputing in Sweden (NAISS) at Alvis partially funded by the Swedish Research Council through grant agreement no. 2022-06725, and by the Berzelius resources provided by the Knut and Alice Wallenberg Foundation at the National Supercomputer Centre.

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

## A  Generating predictions for ParaRel* with LLaMA and Atlas

To evaluate LLaMA on ParaRel*, we compute the log likelihoods of all answer options given a relation, template and subject, and select the answer with the highest likelihood score.

Since Atlas is a seq2seq model, we cannot use the same technique to restrict the prediction candidates as for the BERT-like models. Instead, we make use of an approach similar to the one used for LLaMA. For a given query, we get the latent representation from the Atlas reader encoder and give this as input to the Atlas reader decoder together with each answer option. The answer options are formatted as e.g. "<extra_id_0>France" to suit the T5 format. We then select the answer option with the highest likelihood score according to the reader decoder. To ascertain that the constrained decoding approach is working, we compare the freely generated model answers to the constrained model answers and ensure that they match when the model happens to freely generate an accepted answer option.

## B  Interventions on Retrieval Augmentation

To analyse the effects of retrieval consistency, we develop several strategies for setting consistent retrieved passages: relevant, irrelevant but cohesive, irrelevant and incohesive. These refer to whether the we expect the information in the passages to be related to the questioned asked (relevant) or to the type of relation in the question (cohesive). Table 5 shows a sketched example of how these are assigned.

## C  Measuring reader dependence on retrieval result

We measure reader dependence on the retrieval result by taking the frequency-based rank of the reader prediction compared to all other possible answer alternatives. This is done in two steps: 1) we order the ParaRel* answer alternatives by frequency for each retrieved result and 2) estimate the ranking (normalised between 0 and 1) of the prediction within this order. To relate this metric to consistency, we make the estimates on prompt-pair level by taking the average of the two corresponding prediction rankings. This allows us to estimate both mean rank per relation, but also mean rank for consistent and inconsistent predictions separately.

To give an example how the metric works, consider the input query "Y is located in [X]." and assume that we have the answer candidates *Canada*, *Norway* and *Singapore*. We generate the corresponding Atlas retrieval result with term frequencies as indicated in Table 6 and pass the passages to the reader to get the prediction Y=*Singapore*. According to the term frequencies, this prediction has rank 2, while *Canada* has rank 3 and *Norway* 1. After normalisation to values between 0 and 1, the rank of $Y$ is 0.5 and this is what we report.

## D  Knowledgeable Consistency

ParaRel also proposes two metrics for deeper analysis of cases when the model is expected to know the fact and when the fact may not be in the model. They define *knowledgeable consistency* as the pairwise consistency for any fact, for which any of the prompts was able to retrieve the correct answer; and *unknowledgeable consistency* – pairwise consistency for fact that no prompt is successful. We propose one more metric, which expands on the idea of *knowledgeable consistency*. Since the original metric in ParaRel counts any pair matches between prompts, as long as one of them was correct, this can be misleading in estimating how consistent to the true fact the model is. We define the new metric as the pairwise consistency of prompts, where both prompts agree and are both correct.

## E  Additional results

We calculate the extended metrics described in Appendix D for the models we evaluate (See Table 7). We observe similar trends in performance between the models. One trend we notice is that the *unknowledgeable consistency* plateaus for larger model sizes.

Furthermore, we perform stratified measurement of consistency of retrieval passages depending on wheter the overall model prediction matched (was consistent) or not (See Table 8).

Finally, we perform analysis of our proposed metrics for measuring retrieval consistency and calculate how well they correlate with each other (See Table 9). Unsurprisingly, we find a high correlation between id and title match (since id is sub-level of title). We also find embedding distance correlates with id and title.

| query | Set of retrieved passages | | | |
| --- | --- | --- | --- | --- |
| | no intervention | relevant | irr cohesive | irr incohesive |
| Eibenstock is located in [Y] . | $\{P_1, P_2, P_3\}$ | $\{P_1, P_2, P_3\}$ | $\{P_9, P_{10}, P_{11}\}$ | $\{P_3, P_{10}, P_{14}\}$ |
| Eibenstock, which is located in [Y]. | $\{P_1, P_3, P_4\}$ | $\{P_1, P_2, P_3\}$ | $\{P_9, P_{10}, P_{11}\}$ | $\{P_3, P_{10}, P_{14}\}$ |
| Eibenstock, located in [Y]. | $\{P_2, P_3, P_5\}$ | $\{P_1, P_2, P_3\}$ | $\{P_9, P_{10}, P_{11}\}$ | $\{P_3, P_{10}, P_{14}\}$ |
| Glencree is located in [Y] . | $\{P_9, P_{10}, P_{11}\}$ | | | |
| Robert Bunsen specializes in [Y] | $\{P_{12}, P_{13}, P_{14}\}$ | | | |

Table 5: An example of intervention on the retrieved passages when querying for the location of *Eibenstock*. We refer to the passages as $P_1$, $P_2$, $P_3$... and assume that 3 passages are retrieved for each query.

| Candidate | Frequency |
| --- | --- |
| Canada | 2 |
| Norway | 7 |
| Singapore | 5 |

Table 6: Candidate term frequencies for a retrieval result, as an example.

## F   Duplicated entries in ParaRel

See Table 10 for statistics for the duplicated data entries in ParaRel.

## G   Detailed results for evaluation task format effects on consistency

Detailed results for evaluation task format related effects can be found in Tables 11 to 14, where we report consistency per model and per group.

## H   ParaRel* flags indicating potential data issues

Some queries are identified to have semantically similar answer options, which may cause the model to be unfairly punished for predicting, for example, *science* instead of *biology*. For a summary see Table 15.

Some queries are observed to result in unidiomatic language due to the template structure. For details see Table 16.

Furthermore, resulting texts may be unidiomatic due to the gold label that we want to predict. Detailed list of those objects can be found in Table 17.

Finally, Table 18 shows the full list of relations that contain some overlap between the subject in the prompt and the gold label we want to predict, potentially allowing the model to use shallow heuristics for its prediction.

| Model | Know Cons | K-know Cons | Unk Cons |
|---|---|---|---|
| atlas-base | $0.75 \pm 0.15$ | $0.71 \pm 0.17$ | $0.59 \pm 0.19$ |
| atlas-large | $0.78 \pm 0.14$ | $0.74 \pm 0.15$ | $0.61 \pm 0.20$ |
| atlas-base* | $0.65 \pm 0.24$ | $0.57 \pm 0.28$ | $0.51 \pm 0.20$ |
| t5-base | $0.64 \pm 0.23$ | $0.56 \pm 0.26$ | $0.52 \pm 0.23$ |
| bert-base | $0.63 \pm 0.25$ | $0.56 \pm 0.29$ | $0.45 \pm 0.21$ |
| bert-large | $0.64 \pm 0.24$ | $0.57 \pm 0.28$ | $0.47 \pm 0.20$ |
| llama-7b | $0.69 \pm 0.18$ | $0.63 \pm 0.22$ | $0.48 \pm 0.16$ |
| llama-13b | $0.71 \pm 0.17$ | $0.65 \pm 0.20$ | $0.50 \pm 0.17$ |
| llama-30b | $0.72 \pm 0.17$ | $0.67 \pm 0.19$ | $0.50 \pm 0.17$ |
| llama-65b | $0.73 \pm 0.16$ | $0.69 \pm 0.18$ | $0.50 \pm 0.17$ |

Table 7: ParaRel* results of Atlas, LLaMA and some baselines averaged over all 30 N-1 relations with extra metrics. *=closed-book.

| Model | Metric | Match sim. | No match sim. |
|---|---|---|---|
| atlas-base | *id* | $0.53 \pm 0.10$ | $0.45 \pm 0.11$ |
| | *title* | $0.60 \pm 0.09$ | $0.52 \pm 0.11$ |
| | *embedding* | $0.89 \pm 0.04$ | $0.87 \pm 0.04$ |
| atlas-large | *id* | $0.54 \pm 0.09$ | $0.46 \pm 0.11$ |
| | *title* | $0.60 \pm 0.09$ | $0.52 \pm 0.11$ |
| | *embedding* | $0.89 \pm 0.04$ | $0.87 \pm 0.04$ |

Table 8: The pairwise retriever similarity results stratified over whether the corresponding prediction made by the retriever was consistent. Complement to Table 2. Match consistency refers to the retriever consistency for cases for which the final model prediction is consistent and no match consistency refers to cases for which the final model prediction is inconsistent.

| | id | title | embedding |
|---|---|---|---|
| id | $1.00 \pm 0.00$ | $0.82 \pm 0.20$ | $0.20 \pm 0.12$ |
| title | $0.82 \pm 0.20$ | $1.00 \pm 0.00$ | $0.21 \pm 0.11$ |
| embedding | $0.20 \pm 0.12$ | $0.21 \pm 0.11$ | $1.00 \pm 0.00$ |

Table 9: The correlations between the retriever consistency metrics for random sampling across relations and subjects ('all') averaged over ParaRel* relations. 1000 pairs were sampled per sampling method.

| Relation | Name | #entries | #duplicates | #exact |
|---|---|---|---|---|
| P17 | located-in | 912 | 2 | 0 |
| P19 | born-in | 779 | 0 | 0 |
| P20 | died-in | 817 | 0 | 0 |
| P27 | citizen-of | 958 | 0 | 0 |
| P30 | located-in-continent | 959 | 4 | 0 |
| P36 | capital-of | 471 | 14 | 1 |
| P37 | official-language | 900 | 280 | 0 |
| P101 | specializes-in | 571 | 52 | 0 |
| P103 | native-language | 919 | 2 | 0 |
| P106 | is-a-by-profession | 821 | 0 | 0 |
| P127 | owned-by | 616 | 0 | 0 |
| P131 | located-in | 775 | 0 | 0 |
| P136 | plays-music | 859 | 2 | 0 |
| P138 | named-after | 461 | 23 | 2 |
| P140 | affiliated-with-religion | 432 | 10 | 0 |
| P159 | headquarter-in | 801 | 4 | 0 |
| P176 | produced-by | 925 | 19 | 8 |
| P178 | developed-by | 588 | 12 | 1 |
| P264 | represented-by-music-label | 53 | 2 | 0 |
| P276 | located-in | 764 | 74 | 1 |
| P279 | subclass-of | 900 | 4 | 0 |
| P361 | part-of | 746 | 64 | 0 |
| P364 | original-language | 756 | 6 | 0 |
| P407 | written-in-language | 857 | 31 | 0 |
| P413 | plays-in-position | 952 | 0 | 0 |
| P449 | originally-aired-on | 801 | 9 | 1 |
| P495 | created-in | 905 | 2 | 0 |
| P740 | founded-in | 843 | 0 | 0 |
| P937 | worked-in | 853 | 21 | 0 |
| P1376 | capital-of | 179 | 8 | 1 |
| P1412 | communicated-in | 924 | 2 | 0 |
| Total | | 23097 | 647 | 15 |

Table 10: We count the number of data tuples for each of the 31 ParaRel N-1 relations and indicate how many of these are duplicates with respect to subject and relation. We also indicate how many of these duplicates are exact duplicates, in the sense that a subject-relation-object tuple occurs multiple times for a relation.

| Model | subject-object similarity | no subj-obj similarity |
|---|---|---|
| atlas-base | $0.83 \pm 0.13$ | $0.75 \pm 0.15$ |
| atlas-large | $0.88 \pm 0.10$ | $0.79 \pm 0.13$ |
| bert-base-cased | $0.79 \pm 0.14$ | $0.50 \pm 0.16$ |
| bert-large-cased | $0.81 \pm 0.11$ | $0.55 \pm 0.14$ |
| llama-07b | $0.87 \pm 0.08$ | $0.72 \pm 0.11$ |
| llama-65b | $0.86 \pm 0.11$ | $0.76 \pm 0.10$ |

Table 11: Pairwise consistency for datapoints containing subject-object similarity and for those that do not. Results are based on 9 relations that contain at least 20% subject-object similarity.

| Model | object issue | no object issue |
|---|---|---|
| atlas-base | $0.60 \pm 0.22$ | $0.74 \pm 0.16$ |
| atlas-large | $0.55 \pm 0.23$ | $0.71 \pm 0.16$ |
| bert-base-cased | $0.39 \pm 0.26$ | $0.58 \pm 0.40$ |
| bert-large-cased | $0.45 \pm 0.27$ | $0.59 \pm 0.35$ |
| llama-07b | $0.45 \pm 0.13$ | $0.66 \pm 0.21$ |
| llama-65b | $0.43 \pm 0.14$ | $0.68 \pm 0.20$ |

Table 12: Pairwise consistency for datapoints containing object level issues and for those that do not. Results are based on 3 relations that have been labeled with object issues.

| Model | template issue both | template issue one | template issue none |
|---|---|---|---|
| atlas-base | $0.43 \pm 0.71$ | $0.68 \pm 0.13$ | $0.83 \pm 0.10$ |
| atlas-large | $0.49 \pm 0.73$ | $0.73 \pm 0.13$ | $0.84 \pm 0.03$ |
| bert-base | $0.30 \pm 0.69$ | $0.43 \pm 0.28$ | $0.65 \pm 0.23$ |
| bert-large | $0.32 \pm 0.70$ | $0.48 \pm 0.26$ | $0.72 \pm 0.21$ |
| llama-07b | $0.35 \pm 0.68$ | $0.56 \pm 0.20$ | $0.76 \pm 0.18$ |
| llama-65b | $0.39 \pm 0.69$ | $0.62 \pm 0.17$ | $0.77 \pm 0.18$ |

Table 13: Pairwise consistency for cases where one, both or none of the compared templates have a template issue. Results are based on 6 relations, where we identified template issues.

| Model | semantic overlap | no semantic overlap |
|---|---|---|
| atlas-base | $0.73 \pm 0.13$ | $0.75 \pm 0.17$ |
| atlas-large | $0.78 \pm 0.10$ | $0.77 \pm 0.16$ |
| bert-base-cased | $0.48 \pm 0.23$ | $0.65 \pm 0.23$ |
| bert-large-cased | $0.51 \pm 0.20$ | $0.67 \pm 0.23$ |
| llama-07b | $0.63 \pm 0.17$ | $0.69 \pm 0.19$ |
| llama-65b | $0.67 \pm 0.16$ | $0.74 \pm 0.15$ |

Table 14: ParaRel* results of Atlas and some baselines averaged over all 30 N-1 relations divided into relations that have semantic ovelap in the answer options (12 relations) and those that do not have an overlap (18 relations).

| Relation | Name | Comment |
|---|---|---|
| P19 | born-in | geographic (City/Country) |
| P20 | died-in | geographic (City/Country) |
| P101 | specializes-in | several general options (e.g. art and science) |
| P106 | is-a-by-profession | someone could be multiple roles - e.g a diplomat and politician |
| P131 | located-in | geographic (City/Country) |
| P140 | affiliated-with-religion | includes (Catholicism, Christian, Christianity) and (Islam, Muslim) |
| P159 | headquarter-in | geographic (City/Country) |
| P276 | located-in | geographic (City/Country) |
| P279 | subclass-of | several general options (e.g. art and science) |
| P361 | part-of | several general options (e.g. art and science) |
| P740 | founded-in | geographic (City/Country) |
| P937 | worked-in | geographic (City/Country) |

Table 15: Relations where answer alternatives could have ambiguity due to semantic overlap.

| Relation | Template | Comment |
|---|---|---|
| P19 born-in | [X] is native to [Y]. | e.g. "Claude Arrieu is native to Paris" |
|  | [X] was native to [Y]. | e.g. "Claude Arrieu was native to Paris" |
| P20 died-in | [X] died at [Y]. | preposition |
|  | [X] passed away at [Y]. | preposition |
|  | [X] lost their life at [Y]. | preposition |
|  | [X] succumbed at [Y]. | preposition |
| P27 citizen-of | [X] is [Y] citizen. | [X] is France citizen ->a French citizen |
| P106 is-a-by-profession | [X] works as [Y]. | may require a/an |
|  | [X], who works as [Y]. | may require a/an |
|  | [X]'s occupation is [Y] | may require a/an |
|  | the occupation of [X] is [Y]. | may require a/an |
|  | the profession of [X] is [Y]. | may require a/an |
| P138 named-after | [X] is named in [Y]'s honor. | apostrophe |
|  | [X] was named in [Y]'s honor. | apostrophe |
|  | [X], named in [Y]'s honor. | apostrophe |
|  | [X], which is named in [Y]'s honor. | apostrophe |
|  | [X], which was named in [Y]'s honor. | apostrophe |
| P1376 capital-of | [Y]'s capital, [X]. | apostrophe |
|  | [Y]'s capital city, [X]. | apostrophe |
|  | [Y]'s capital is [X]. | apostrophe |
|  | [Y]'s capital city is [X]. | apostrophe |

Table 16: Relations containing unidiomatic templates.

| P101 - specializes in | P138 named after | P361 part of | | | |
|---|---|---|---|---|---|
| Internet | Alps | Alps | car | foot | perfume |
| astronomer | Americas | Americas | cartridge | forest | pistol |
| bird | Arctic | Antarctic | castle | fruit | piston |
| car | Bible | BBC | cavalry | galaxy | port |
| cave | Moon | Bible | cell | gang | radar |
| comedian | Netherlands | Caribbean | cemetery | gene | saddle |
| diplomat | Sun | Caucasus | chromosome | genome | screw |
| economist | arrow | Internet | clergy | gospel | sea |
| habitat | backpack | Nile | cloud | graph | seed |
| hotel | brake | Quran | cocktail | head | shield |
| icon | canon | airline | coin | heart | skeleton |
| mathematician | cube | airport | comet | kidney | skull |
| miniature | flower | ankle | computer | leaf | spacecraft |
| musical | glove | aquarium | door | liver | stomach |
| musician | grape | army | ear | lung | sword |
| nightclub | horse | artillery | economist | matrix | track |
| novelist | hotel | atom | ecosystem | molecule | trail |
| philosopher | liver | banana | engine | mosque | tree |
| physician | mayor | battery | enzyme | municipality | triangle |
| physicist | mole | bicycle | eye | navy | turbine |
| priest | monastery | bird | facade | neck | volcano |
| programmer | patent | bow | film | nerve | |
| stock | patriarch | brain | firearm | orbit | |
| stomach | red | breast | fish | organism | |
| virus | | bridge | fleet | parish | |
| website | | candle | flower | penis | |

Table 17: Relations containing unidiomatic objects.

| Relation | Name |
|----------|------|
| P36 | capital-of |
| P127 | owned-by |
| P131 | located-in |
| P138 | named-after |
| P176 | produced-by |
| P178 | developed-by |
| P276 | located-in |
| P279 | subclass-of |
| P361 | part-of |

Table 18: Relations containing subject and object similarity.