# OpenReview forum: "The Effect of Scaling, Retrieval Augmentation and Form on the Factual Consistency of Language Models"
_EMNLP/2023/Conference — EMNLP 2023 Main_

### Official Review · Reviewer_Moz1 · 2023-07-29

**Soundness:** 3

**Excitement:**

4: Strong: This paper deepens the understanding of some phenomenon or lowers the barriers to an existing research direction.

**Missing References:**

N/A

**Paper Topic And Main Contributions:**

This paper presents a set of experiments on the effect of scaling, retrieval augmentation and form on 'consistency' of language models, that is, the ability of a model to deliver consistent answers to semantically equivalent questions. To do so, the authors use the ParaRel dataset developed for precisely this problem (consistency) which consists in a set of semantically equivalent cloze-style English query paraphrases for 'triple' relations. The dataset is first cleaned by the authors, removing duplicates.

The authors find that up-scaling (using bigger models) somewhat improves consistency but not as much as retrieval augmentation (using the RAG model Atlas) which has a more significant effect on consistency. 'Form' which is a combination of different issues, specific to the dataset, is also shown to have an effect. The authors also investigate whether retrieved text similarity has an effect on consistency (it has), and whether using same retrieved text for all equivalent queries has an effect on response consistency (it has if it is relevant), amongst other experiments.

**Questions For The Authors:**

See reasons to reject above (as QA, QB, QC, and QD)

**Reasons To Accept:**

A multi-experiment study focused on ways to improve consistency in LLMs and the identification of elements that affect consistency, using the ParaRel benchmark dataset.

**Reasons To Reject:**

A) The paper uses the ParaRel dataset which addresses consistency in a very specific task, which I feel should be reflected in the title of the paper. Else, the paper promises in its title more than it can deliver.
B) Some aspects/hypthesis of the study are very specific to (and are made to fit) the ParaRel dataset and quite idiosyncratic: I am thinking of the experiments on the effect of Form, whereby Form is taken to include an heterogeneous set of 'phenomena' such as use of incorrect preposition, some minor errors for unidiomatic objects, semantic overlap (in what way is this 'form'?).
C) I found the paper quite a hard read, requiring a lot of long term memory :-D and shuffling backwards and forwards, as all the settings and hypothesis of the (many) experiments are presented first and then the results are presented afterwards. I would prefer a different organization of the paper whereas each group of hypothesis/experiments is explained, and results of the experiment are shown immediately thereafter.
D) The paper lacks an element of 'excitement' (it seems hardly surprising that generating a response given relevant retrieved documents will produce better results, or that the more parameters a model have, the better the results will be). Also the paper lacks vision outside the ParaRel experiment. The authors make some hints that things might be more complicated and not everything is explained by what they study, but it is very much left to that.

**Reproducibility:**

4: Could mostly reproduce the results, but there may be some variation because of sample variance or minor variations in their interpretation of the protocol or method.

**Reviewer Confidence:**

3: Pretty sure, but there's a chance I missed something. Although I have a good feel for this area in general, I did not carefully check the paper's details, e.g., the math, experimental design, or novelty.

**Typos Grammar Style And Presentation Improvements:**

As mentioned in reasons to reject, a reoganization of the paper might improve its readability.
Also, providing some examples might be helpful at some points of the paper. For example the procedure explained in lines 450 to 462.

---

> ### Author Rebuttal · Authors · 2023-08-28
>
> Thank you for your review. We have mainly based our rebuttal on your _Reasons To Reject_ and _Questions For The Authors_. Please let us know if there is anything we should address in addition to this.
>
> > A) The paper uses the ParaRel dataset which addresses consistency in a very specific task, which I feel should be reflected in the title of the paper. Else, the paper promises in its title more than it can deliver.
>
> Fair point, we can revise the title. Would for example the title “The Effect of Scaling, Retrieval Augmentation and Form on the Factual Consistency of Language Models” be acceptable?
>
> > B) Some aspects/hypothesis of the study are very specific to (and are made to fit) the ParaRel dataset and quite idiosyncratic: I am thinking of the experiments on the effect of Form, whereby Form is taken to include an heterogeneous set of 'phenomena' such as use of incorrect preposition, some minor errors for unidiomatic objects, semantic overlap (in what way is this 'form'?).
>
> Agreed, our study is largely adapted to the use of ParaRel for measuring consistency and it could be argued that the query related effects we observe on consistency only are relevant to datasets created by automatic template based generation. However, ParaRel is not the only dataset to be created in this way (and it is probably not the last), so measuring the effect on factual consistency of slightly imperfect language usage should still be of value, e.g. when arguing for why ParaRel is not the best approach for measuring factual consistency and why we need better evaluation methods. Additionally, we do not know in what settings LMs will be used in the future and we cannot rule out that we might require our models to remain factually consistent regardless of whether a question is expressed as “Did Napoleon die in Paris?” or “Did Napoleon die at Paris?”, which could motivate an investigation of such effects of imperfect language usage.
>
> About our use of the word “form” for a wide array of phenomena. We are aware that the term is used a bit broadly and will try to refine the terminology and/or add a discussion on how those groups fit under the same category of phenomena.
>
> > C) I found the paper quite a hard read, requiring a lot of long term memory :-D and shuffling backwards and forwards, as all the settings and hypotheses of the (many) experiments are presented first and then the results are presented afterwards. I would prefer a different organization of the paper whereas each group of hypothesis/experiments is explained, and results of the experiment are shown immediately thereafter.
>
> We understand the point of the reviewer, and would certainly consider reordering parts of the paper, should all reviewers agree this is preferable. If there were specific sections or experiments that were hard to follow, we would also appreciate it if you could point these out, alternatively let us know if the issue lies with the whole structure.
>
> Also, if you have any examples of papers that followed your suggested setup in a nice way, we would be happy to receive them as inspiration.
>
> > D) The paper lacks an element of 'excitement' (it seems hardly surprising that generating a response given relevant retrieved documents will produce better results, or that the more parameters a model have, the better the results will be).
>
> This comment is similar to one of Reviewer 91nZ’s so please also refer to our answer there. To summarize, we would like to argue that 1) just because something may be considered expected it should not mean that it should not be supported by empirical evidence and 2) we do more than to simply investigate whether scaling or retrieval augmentation improves on consistency. For example, we:
> 1. perform a relative comparison of scaling vs. retrieval-augmentation,
> 2. do a causal analysis on Atlas to find that providing perfectly consistent passages across query paraphrases does not lead to perfect consistency, indicating that retrieval-augmentation is not the full story for consistency, and
> 3. find that scaling LLaMA has diminishing returns with respect to consistency (scaling from 30B to 65B parameters yields insignificant consistency improvements) and is therefore not the full solution.
>
> Points two and three above should be of special interest, given the reviewer's claim that those augmentations should yield expected, and therefore unexciting, improvements. (2) shows us that perfect retrieval-augmentation only marginally improves consistency and (3) clearly contradicts the assumption that the more parameters a model has, the better the results will be.
>
> > Also the paper lacks vision outside the ParaRel experiment. The authors make some hints that things might be more complicated and not everything is explained by what they study, but it is very much left to that.
>
> We agree and will expand on the Future Work section. Some examples on potential directions, mostly focused on exploring new model types, were given in the response to Reviewer 91nZ. Another direction we are interested in is to further investigate the interaction between reader and retriever for retrieval augmented models, for example when there are conflicts between retrieval corpus and reader weights, similarly to the work on “Entity-Based Knowledge Conflicts in Question Answering” by Longpre et al. (2021). This might explain why the reader could be inconsistent in spite of being provided with perfectly consistent passages, as observed in our paper. It would also be interesting to investigate the effect of the interaction between reader and retriever during training, for example if the reader to some extent memorizes less information as a result of receiving passages with the information, where one could use a method similar to “Locating and Editing Factual Associations in GPT” by Meng et al. (2023). Lastly, improved consistency and retrieval augmentation could highlight a path towards more explainable LMs, holding the promise of models that are explainable by design.

---

### Official Review · Reviewer_Urmf · 2023-08-01

**Soundness:** 3

**Excitement:**

4: Strong: This paper deepens the understanding of some phenomenon or lowers the barriers to an existing research direction.

**Paper Topic And Main Contributions:**

This paper studies the impact of two different strategies to mitigate inconsistencies during access to factual knowledge from LLMs, where such inconsistencies manifest in the form of variability in the results produced by semantically equivalent queries to the same model. The first strategy focuses on simply upscaling the model, i.e. replacing a given model with other versions of such model with an increasing parameter count. To this purpose, the authors use the LLaMA family of LLMs. The second strategy involves augmenting the LLM with information retrieved from a collection of documents, through a retrieval-reader architecture model like Atlas, which is used for experimentation and compared to non-retrieval models of similar scale.

The results show that although both strategies contribute to improve consistency, the latter does so with larger gains and more efficiently so. However interesting this result may be, the main contribution of the paper IMO is the extensive analysis carried out to drill down on the actual reasons behind this finding. To this purpose, the authors f thocus on the ParaRel benchmark, derived from Lama, adapting it for the task at hand. Such experimentation deal with the possible factors related to consistency along the following dimensions: consistency of the retrieval, correlation between retrieval consistency and prediction consistency, consistency and relevance of the retriever vs. consistency and accuracy of the predictions, and dependency of the reader on term frequency in the retrieved results.

**Reasons To Accept:**

+ The study of consistency in tasks related to factual knowledge has been limited compared to accuracy so far. This work is timely in that direction.
+ The paper is well structured and its narrative, understood as the coherence between the research hypoteheses and the experimentation, flows naturally.
+ The approach is well formalized.
+ Experimentation on the ParaRel becnhmark is well-deisgned, abundant and offers convincing results.

**Reasons To Reject:**

- I would have like to see if the results obtained from the experimentation generalize to more datasets.
- I would have also liked to see a retriever-reader baseline build on LLaMA or similar LLM to see how the combination of higher parameter count and retrieval-based enhancement impact the overall results

**Reproducibility:**

4: Could mostly reproduce the results, but there may be some variation because of sample variance or minor variations in their interpretation of the protocol or method.

**Reviewer Confidence:**

4: Quite sure. I tried to check the important points carefully. It's unlikely, though conceivable, that I missed something that should affect my ratings.

---

> ### Author Rebuttal · Authors · 2023-08-28
>
> Thank you for your review. We have mainly based our rebuttal on your _Reasons To Reject_. Please let us know if there is anything we should address in addition to this.
>
> > I would have liked to see if the results obtained from the experimentation generalize to more datasets.
>
> As would we. But as mentioned in our response to reviewer 91nZ, we know of no other datasets that could help us measure that kind of generalizability with respect to factual consistency. To our knowledge, ParaRel is the largest and most relevant dataset for these types of evaluations. If you have recommendations on additional datasets to consider, we would be happy to receive them.
>
> > I would have also liked to see a retriever-reader baseline build on LLaMA or similar LLM to see how the combination of higher parameter count and retrieval-based enhancement impact the overall results.
>
> That is definitely interesting future work. But we would firstly like to reiterate that the main goal of our paper is not to chase the new SoTA on ParaRel. Our focus lies on measuring and disentangling effects on consistency such that we can understand underlying mechanisms.
>
> Secondly, to develop a “fair” representation of a retrieval-augmented LLaMA it would be necessary to find a suitable information fusion method and train the reader to learn to use the retrieved information, similarly to how Izacard et al. (2022) developed Atlas. To successfully develop such a reader-retriever network is not a simple feat and should amount to its own paper, as was the case for Atlas. However, we would be happy to add this under a future work section.
>
> To expand on our answer above, we would like to point out that even if we were to use the simplest possible approach for a retrieval-augmented LLaMA, there would be significant and time-consuming choices regarding the method design that would need to be evaluated. We could for example take the passages that were retrieved by Atlas and give them to LLaMA to form a baseline, but it would require figuring out a good way of doing the information fusion for 20 or so passages. Atlas uses FiD but LLaMA cannot do that since it does not build on the T5 model, so one would need to find an alternative approach for fusing large quantities of information into the reader. We could simply append as many retrieved passages as possible to the query (similar to the REALM setup by Guu et al. (2020)) but this is probably not SoTA. To conclude, we would either get something underdeveloped method-wise or need to write a new paper.

---

### Official Review · Reviewer_91nZ · 2023-08-04

**Soundness:** 3

**Excitement:**

3: Ambivalent: It has merits (e.g., it reports state-of-the-art results, the idea is nice), but there are key weaknesses (e.g., it describes incremental work), and it can significantly benefit from another round of revision. However, I won't object to accepting it if my co-reviewers champion it.

**Paper Topic And Main Contributions:**

The paper discusses two methodologies for addressing challenges of large language models with regards to inconsistencies: up-scaling the LLM, and the usage of retrieval databases.

The two approaches are evaluated with an improved version of the ParaRel benchmark. The results show that retrieval augmentation shows better performance compared to the up-scaling of LLMs, and is also more efficient.

The authors then further analyse the retrieval augmentation approach, e.g. with regards to metrics for measuring retrieval consistency.

The authors conclude that even with the retrieval augmentation approach, there is no full consistency.





**Questions For The Authors:**

The authors conclude in section 6 that

"Retrieval augmentation is discovered as the best method for consistency improvement, while it does not achieve perfect consistency."

Given this statement, it would be interesting to learn what other approaches the authors could foresee which could provide perfect consistency. One area to explore would be the usage of knowledge graphs as a source of explicit factual knowledge. One could expect that a given query to the same version of the graph will allows provide the same results.

**Reasons To Accept:**

The paper empirically analysis the effect of different methods on consistency of LLMs.

**Reasons To Reject:**

The results that retrieval augmentation improves performance is not much of a surprise. The paper confirms this not with accuracy but with consistency as a benchmark criterion. In that way, the paper does not go far beyond the state-of-the-art, but mostly improves an existing benchmark about consistency (ParaRel).

**Reproducibility:**

4: Could mostly reproduce the results, but there may be some variation because of sample variance or minor variations in their interpretation of the protocol or method.

**Reviewer Confidence:**

3: Pretty sure, but there's a chance I missed something. Although I have a good feel for this area in general, I did not carefully check the paper's details, e.g., the math, experimental design, or novelty.

---

> ### Author Rebuttal · Authors · 2023-08-28
>
> Thank you for your review. We have mainly based our rebuttal on your _Reasons To Reject_ and _Questions For The Authors_. Please let us know if there is anything we should address in addition to this.
>
> > The results that retrieval augmentation improves performance is not much of a surprise. The paper confirms this not with accuracy but with consistency as a benchmark criterion.
>
> It is correct that it has already been proven that retrieval augmentation can improve _accuracy_ on several QA and NLU tasks. However, to the extent of our knowledge, there are no preceding published results that have found retrieval augmentation to improve on consistency or otherwise similar robustness metrics. While the reviewer may not find this surprising, such claims still ought to be empirically validated.
>
> Additionally, we would like to argue that the main contributions of our paper are not only that we establish retrieval augmentation as a way to improve consistency. We move on to compare the strength of the effect of retrieval augmentation against other approaches that can be expected to perform equally well, such as scaling, and show that retrieval-augmentation outperforms scaling consistency-wise. We also investigate potential reasons for _why_ retrieval augmentation improves on consistency and why it does not achieve perfect consistency, considering e.g. retriever consistency and unidiomatic language usage.
>
> > In that way, the paper does not go far beyond the state-of-the-art, but mostly improves an existing benchmark about consistency (ParaRel).
>
> We did not intend for our contributions to be to obtain state-of-the-art on general benchmarks that measure accuracy, or to improve benchmark results on ParaRel. Our intended contribution lies in diagnosing the factual consistency of modern LMs, for which we use ParaRel as the measurement tool. To our knowledge, there are no other existing datasets that can be used to measure factual consistency or similarly related robustness metrics for foundational LMs. If you know of any additional benchmarks suitable for consistency evaluations, we would be happy to hear of them.
>
> > Given this statement, it would be interesting to learn what other approaches the authors could foresee which could provide perfect consistency. One area to explore would be the usage of knowledge graphs as a source of explicit factual knowledge. One could expect that a given query to the same version of the graph will provide the same results.
>
> This answer will have to be a pure speculation since we do not currently have empirical results to back up any claims. It is on the agenda to consider models that build on knowledge graphs, such as TIARA by Shu et al. (2022), but it remains to be seen whether they are more or less consistent (and accurate) than models based on text retrieval and we have no prior opinion on the outcome of such a comparison. Just as we have investigated in this paper for text retrieval, the accuracy and consistency of KG-based models will depend on the robustness of query representations under paraphrasing and how sensitive the retrieval component is to such variation.
>
> The goal of the current paper was not to modify any existing system to achieve improved consistency, but rather to investigate the relative strengths and interplay of various factors that influence consistency in LMs. Focusing on the task of improving consistency, we do not expect any approach to achieve 100% consistency given the current state of the field. Apart from changing model setup, one potential avenue to investigate that would take us closer to perfect consistency is to consider model confidences, since there are probably situations for which a model does not “know” the necessary fact and therefore performs random guessing from the start. We expect that providing the models with the ability to skip when blank for a certain fact or phrasing should improve on consistency. TIARA has this ability for example.

---

### Meta-Review · Area_Chair_bTkG · 2023-09-18

**Recommendation:** 5

**Metareview:**

The paper studies the inconsistencies problem in LLMs. It compares two methodologies: up-scaling the LLM, and the usage of retrieval databases. With an improved version of the ParaRel benchmark the results show that retrieval augmentation performs better and is more efficient. However, even with the retrieval augmentation approach, there is no full consistency. extensive analysis drills down on the reasons behind this finding along various dimensions: consistency of the retrieval, correlation between retrieval consistency and prediction consistency, dependency of the reader on term frequency in the retrieved results, etc.

Strength:
1. Introduce an  improved benchmark (ParaRel) by considering consistency.
2. The approach is well formalized.
3. The result  is  abundant and convincing.

Weakness:
1. The results that retrieval augmentation improves performance is not much of a surprise.
2. The result is restricted to only one dataset
3. There is no result for the combination of higher parameter count and retrieval-based enhancement

---

### Decision · Program_Chairs · 2023-10-07

**Decision:**

Accept-Main

**Comment:**

The paper studies the inconsistencies problem in LLMs. It compares two methodologies: up-scaling the LLM, and the usage of retrieval databases. With an improved version of the ParaRel benchmark the results show that retrieval augmentation performs better and is more efficient. However, even with the retrieval augmentation approach, there is no full consistency. extensive analysis drills down on the reasons behind this finding along various dimensions: consistency of the retrieval, correlation between retrieval consistency and prediction consistency, dependency of the reader on term frequency in the retrieved results, etc.

Strength:
1. Introduce an  improved benchmark (ParaRel) by considering consistency.
2. The approach is well formalized.
3. The result  is  abundant and convincing.

Weakness:
1. The results that retrieval augmentation improves performance is not much of a surprise.
2. The result is restricted to only one dataset
3. There is no result for the combination of higher parameter count and retrieval-based enhancement